# Exploration of Outliers in If-Then Rule-Based Knowledge Bases

**DOI:** 10.3390/e22101096

**Published:** 2020-09-29

**Authors:** Agnieszka Nowak-Brzezińska, Czesław Horyń

**Affiliations:** Institute of Computer Science, Faculty of Science and Technology, University of Silesia, Bankowa 12, 40-007 Katowice, Poland; czeslaw@horyn.pl

**Keywords:** rule-based knowledge base, outliers detection, cluster validity, data clustering, *AHC*, *LOF*, *COF*, *K-MEANS*, *SMALL CLUSTERS*

## Abstract

The article presents both methods of clustering and outlier detection in complex data, such as rule-based knowledge bases. What distinguishes this work from others is, first, the application of clustering algorithms to rules in domain knowledge bases, and secondly, the use of outlier detection algorithms to detect unusual rules in knowledge bases. The aim of the paper is the analysis of using four algorithms for outlier detection in rule-based knowledge bases: Local Outlier Factor (LOF), Connectivity-based Outlier Factor (COF), *K*-MEANS, and SMALL
CLUSTERS. The subject of outlier mining is very important nowadays. Outliers in rules *If-Then* mean unusual rules, which are rare in comparing to others and should be explored by the domain expert as soon as possible. In the research, the authors use the outlier detection methods to find a given number of outliers in rules (1%, 5%, 10%), while in small groups, the number of outliers covers no more than 5% of the rule cluster. Subsequently, the authors analyze which of seven various quality indices, which they use for all rules and after removing selected outliers, improve the quality of rule clusters. In the experimental stage, the authors use six different knowledge bases. The best results (the most often the clusters quality was improved) are achieved for two outlier detection algorithms LOF and COF.

## 1. Introduction

The data analyzed in this work are rule-based knowledge bases (KBs) on which the Decision Support Systems (DSSs) are based, supporting organizational and business decision-making. DSSs need artificial intelligence and machine learning for an effective decision support process. These techniques can not only accelerate the processes of inference but also increase the quality of decisions made through the use of effective learning methods. Many authors attempt to use machine learning methods, e.g., clustering, in the decision support process. The article [1] presents a DSS for diabetes prediction while using machine learning and deep learning techniques. The authors compared conventional machine learning (Support Vector Machine (SVM) and the Random Forest (RF)) with deep learning approaches to predict and detect the diabetes patients. In [2], the authors present a clinical DSS for brest cancer screening using clustering and classification in which the Partition Around Medoid (PAM) clustering algorithm is used. In turn, in [3], the data-driven weather forecasting models performance comparison for improving offshore wind turbine availability and maintenance was presented by the authors. It is based on two approaches long short-term memory network (LSTM), and Markov chain. The analyzed papers present attempts to use machine learning methods to support decision-making; however, it was not possible to find papers that would approach the decision-making support, like the authors of this paper. We propose using rule clustering to implement the inference processes, which are the heart of each DSS, faster and more efficiently. For this purpose, we use hierarchical rule agglomeration algorithms (other than the clustering algorithms used in the analyzed works).

Rule clustering is a way to obtain a large number of rules in the KB that ar necessary for searching when we are looking for the rules to activate. Discovering outliers in the rules is intended to support the knowledge engineer in ensuring the consistency and completeness of the knowledge stored in the KBs. Identifying unusual rules will allow you to take care of the development of the KB in an under-explored scope as soon as possible. Rule clusters are supposed to have a positive effect on the effectiveness of inference; however, taking into account unusual rules that may interfere with the coherence and separability of clusters, it was necessary to investigate what impact actually outliers among the rules have on the quality of the clusters of rules and, thus, on the effectiveness of inference.

Knowledge is one of the key factors in making effective decisions in a wide range of areas of life. The most popular knowledge representation is the one that is based on rules *If-Then*. Therefore, the analyzed data are rules in KBs. An example of a rule in the field of diabetes is: (2−Hour_serum_insulin=0)&(Number_of_times_pregnant=6)&(Plasma_glucose_concentration=125)=>(Class=1[2]), where (2−Hour_serum_insulin=0), (Number_of_times_pregnant=6) where (Plasma_glucose_concentration=125) forms *If* part and (Class=1[2]) forms *Then* part, respectively. This rule format is generated by the Rough Set Exploration System (RSES) [4]. Unusual rules do not create any clusters. Sometimes they create small clusters—clusters with very few rules, for example, no more than 5% of all rules. An unusual are also rules that significantly affect the representation of the whole rule cluster, or deteriorate the quality of clusters. The rule-based KBs, in addition to explicitly expressed domain knowledge, contain additional, hidden knowledge, dispersed in sets of rules of potentially large numbers. This knowledge often reflects the experts’ hard to formalize competences (generalizations and clandestine relationships), intuitive knowledge, exceptions, special cases, and so-called mental shortcuts. Unusual (rare) rules should be analyzed by an expert in a given field. Discovering such type of rules in a certain context gives the opportunity to explore knowledge in a given domain and affects the efficiency of inference in DSS. For scientific fields such as medicine, economics, management, psychology, and many others, this is a very important issue. Effective exploration of KBs is understood as: reducing the time that is required to apply and extracting knowledge that is useful to the user’s expectations (domain expert, knowledge engineer).

Today’s DSSs require large amounts of data collection, rich and consistent knowledge, and are, therefore, most often equipped with large KBs. Big, that is, with a few hundred or a few thousand of rules. Such large KBs are too difficult for knowledge engineers to perform efficient and comprehensive quality monitoring of these data sources. However, this quality must be maintained if we want such systems to replace contact with a domain expert. When there are a lot of rules in the KB, it is necessary to use machine learning techniques that will allow for effective management of large amounts of data. Cluster analysis is one of the techniques. By clustering similar rules into groups and creating representatives for such groups, not only do we learn the internal data structure (hierarchy and numerous connections), but also allow faster data retrieval later in such a structure. When we search such groups, it is enough to compare the information we are looking for with the group representatives to find the group that best fits the information we are looking for. This significantly reduces the analysis time. In principle, the more data are clustered, the later the greater the profit from building clusters and viewing them instead of reviewing each object (rule) separately. When creating rule clusters, it should be born in mind that, if there are unusual rules among the rules, which are not similar to others, they will affect the quality of clusters and the quality of their representatives and, thus, may impair the effectiveness of the inference. In the case of KBs with unusual rules, it would be most desirable to search for all such rules and return them to the domain expert in order to take care of the completeness of the rules and expand knowledge in this rare (incomplete) area. An agglomerative hierarchical clustering (AHC) algorithm has been proposed, which ensures that outliers are included in the clusters at the very end without making the inference process impossible. However, in this paper, it was decided to use algorithms that are known from the literature for outlier mining, like LOF or *K*-MEANS described in Section 3, in order to ascertain the impact of outliers on the quality of the structure of the KB. Additionally, an original approach (SMALL
CLUSTERS) was proposed and the results were verified. The algorithms used different approaches for the issue of finding deviations in the data. The LOF and COF algorithms measure the distances of individual objects from other objects in the group or group representatives. The *K*-MEANS algorithm for a fixed number of clusters (by cutting off the previously created structure at the level of *k* clusters counting from the top) checks which objects stand out the most from the created clusters, and the proprietary approach, the SMALL
CLUSTERS algorithm, indicates all of the clusters with too small numbers as potential outliers. The negative influence of unusual rules on the quality of clusters and, thus, on the efficiency of inference is only a thesis. Using quality assessment measures that are known from the literature, e.g., Dunn’s index, CPCC coefficient, and others, we measure the quality of clusters with outliers and after their isolation, and check for which outlier detection algorithms and which cluster quality assessment measures, the quality of clusters after isolation of outliers has improved the most.

## 2. Rule Clustering

The quality of the KB reflects the quality of the whole DSS. In the case of the rule-based representation of knowledge, the data on the basis of which knowledge is created are usually in the form of a decision table, in which each line consists of a set of conditions and a decision-making part (conclusions). Outliers in the context of the rule-based KBs will be rules that do not match the rest of the knowledge (rules) stored in the KB. Such rules are rare and should be subject to more rigorous research in the field, in particular, by experts, so as to extend the KB to a previously uninvestigated area as far as possible in the future. Detecting outliers in the rules will allow optimizing inference in rule-based KBs and, thus, increase the effectiveness of DSS. The goal of inference process is to find a relevant rule in the whole rule set and to active it. When there is a lot of rules in a given KB (usually we have hundreds or even thousands of rules), this process is time consumming. In order to avoid it a smart organization of rules which allow to find a given rule in shorter time is necessary to develop. One of the method possible to use is rule clustering. When we group similar rules into the groups, especially when we are able to build the hierarchical structure of such groups, the process of searching the KB will be limited to search only a small part of the whole set. The authors considered various types of clustering algorithms however the hierarchical algorithms seem to be an optimal choice. When we use hierarchical clustering algorithms, for example AHC algorithm, the result is the binary tree organization. At the leaves level, we have rules and, going up to the root, we cluster two the most similar rules (and further rule clusters) into a cluster at the higher level. We can stop clustering at the proper level, for example when we reach the desired number of clusters or when the similarity between clustered rules/rule cluster is too small. Having a binary hierarchy, the searching process is organized as follows. Starting from the root or the higher level in the structure, we find rule/rule cluster to be the most relevant to the set of facts (input data). The most relevant means the most similar. The previous research work of the authors showed that, instead of searching whole KB with all rules, we usually only search a small piece of it (the experiments showed that usually only a few percent of the total set was reviewed). More about this can be found in [5,6]. Rule clustering allows for the user to browse the KB faster. However, this is not the only problem that is faced by the knowledge engineer. Very often, when creating a rule-based KB, there is a risk that a certain area of domain knowledge will be incomplete. For example, there will be few rules in a given area that have unusual characteristics in the description. This is very important for two reasons:firstly, many clustering algorithms are then loaded with unusual objects, and they disturb cluster representatives and affect the efficiency of cluster searching; and,secondly, very often the identification of outliers implies that they are omitted from the set to ensure the quality of the clusters.

The existence of outliers in the rules will certainly affect the quality of the created clusters of rules and, thus, the effectiveness of the systems (applications) based on rule cluster representation. Therefore, the first approach is to examine the impact of outliers on the quality of the rule clusters. The second approach is to identify outliers and report them to the domain experts in order to help in expanding the KB that is in a limited (insufficiently explored) scope.

Literature has a variety of work on cluster validity, but they do not concern such specific data as rule-based KBs. The studies used seven different clustering quality indicators in order to examine the quality of the resulting clusters after removing different numbers of unusual rules (1%, 5%, 10%) of data (In the case of SMALL
CLUSTERS algorithm it was assumed that a group containing up to 5% of the set is the so-called small group treated as outliers.) The choice was determined not only by the intuitiveness of these metrics, but also their implementation availability in such popular environments as *R* and Python.

### 2.1. Thesis of Work

The detection of unusual rules (which are different comparing to the rest of rules) before the clustering process improves the quality of the received groups, because unusual rules distort the formed clusters and make data analysis difficult. The goal of the research is to show differences in the quality of the rule clusters when we include and exclude unusual rules (outliers) in the analyzed data set. In order to do this, each experiment includes clustering the rules, including unusual rules, determining the quality of the clusters that are created in this way while using various quality indices, then isolation of selected unusual rules and repeating the clustering and quality assessment procedure for the structure without unusual rules. A comparison of methods for assessing the quality of the clusters in terms of consistency of results is also an important issue in the research. Six different KBs were used in the experimental phase. They differ in both the number of attributes or rules and the type of the data.

### 2.2. Similarity between Rules

There are many different methods to find the similar rules or rule clusters. When a given KB contains data of different type (qualitative, quantitative or binary) the best solution is to use the measures that work well with mixed data type. Such a measure, well known in literature, is Gower distance [7]. Briefly, to compute the Gower distance between two items *x* and *y*, it is necessary to compare each element and compute a term. If the element is numeric, then the term is the absolute value of the difference divided by the range. If the element is non-numeric the term is 1 if the elements are different or the term is 0 if the elements are the same.
(1)$$s\left(x,y\right)=\frac{\sum_{i=1}^{n}s_{i}\left(x,y\right)}{\sum_{i=1}^{n}\delta_{i}}$$For each variable, we calculate the partial similarity coefficient ($s_{i}\left(x,y\right)$) according to the following formula:$s_{i}\left(x,y\right)=1-\frac{|x_{i}-x_{j}|}{\mathrm{the}\;\mathrm{range}\;\mathrm{of}\;i-\mathrm{th}\;\mathrm{variable}}$ if the *i*-th variable is a quantitative attribute$s_{i}\left(x,y\right)=1$ for $x_{i}=y_{i}$ or 0 in opposite case if the *i*-th variable is a quality attribute.$s_{i}\left(x,y\right)=1$ for $x_{i}=1$ and $y_{i}=1$ or 0 for $x_{i}=1$ and $y_{i}=0$ or vice versa ($x_{i}=0$ and $y_{i}=1$) if the *i*-th variable is a binary attribute.

The final distance measure is calculated as the average of the calculated metric, to which the user gives the weights that were determined at his discretion. Details of Gower distance is out of scope of this article and those interested in the details are advised to study [7]. In this research, apart from the Gower distance, the following distance measures were analyzed and used: Euclidean, City, Minkowski, Czebyszew, and Hamming.

### 2.3. Clustering Algorithms

Hierarchical clustering algorithms are classical clustering algorithms where sets of clusters are created. Cluster analysis is a fundamental task in data mining and machine learning area, which aims to separate a bunch of data points into different groups, so that similar points are assigned into the same cluster. The pseudocode of the hierarchical clustering algorithm defined by S.C. Johnson in 1967 is as follows:Step 1: start by assigning each item, from a given set of *N* items, to a single cluster and calculate the distances (similarities) between each pair of the clusters (items).Step 2: find the closest (most similar) pair of clusters and merge them into a single cluster, so that now you have one cluster less.Step 3: compute distances (similarities) between the new cluster and each of the old clusters.Step 4: repeat steps 2 and 3 until all items are clustered into a single cluster of size *N*.

Very often, in case of rule clustering, there is no sense to continue clustering until one cluster with *N* items is achieved. While clustering, the closer to the root of the dendrogram, the less similar the clustered items are. Sometimes, at a certain level, there is no longer any similarity. Therfore, the clustering process should be terminated when the quality of the clusters is the best.

### 2.4. Cluster Validity

The term cluster validity is defined as the process of assessing the reliability of the clustering algorithms while using different initial conditions and as a quantification of the results that were obtained from the clustering. The cluster validity indices are used to identify optimal number of clusters, which provide the effective partitions into homogeneous regions. They evaluate the degree of similarity or dissimilarity between the data and are classified into two major groups, (i) external indices and (ii) internal indices. For validating a partition, external indices compare with the precise partitions while internal indices examine the clustered data set. The most commonly used cluster validity indices Dunn, Davies–Bouldin, cophenetic correlation coefficient (CPCC) and Silhoutte index are described below.

The aim of Dunn index is to identify sets of clusters that are compact, with a small variance between members of the cluster, and well separated, where the means of different clusters are sufficiently far apart, as compared to the within cluster variance. The higher the Dunn index value, the better is the clustering. The number of clusters that maximizes the Dunn index is taken as the optimal number of clusters *k*. It also has some drawbacks. As the number of clusters and dimensionality of the data increase, the computational cost also increases. The Dunn index for *k* number of clusters is defined as Equation (Equation 2):
(2)$$D\left(u\right)=min_{1\leq i\leq k}\left\{min_{1\leq j\leq k,j\neq i}\left\{\frac{\delta (X_{i},X_{j})}{max_{1\leq c\leq k}\left\{\Delta \left(X_{c}\right)\right\}}\right\}\right\}$$
where $\delta (X_{i},X_{j})$ is the inter-cluster distance between cluster $C_{i}$ and $C_{j}$, and $\Delta (X_{c})$ is the intra-cluster distance of cluster $X_{c}$.For Davies–Bouldin index (DBI) the validation of how well the clustering has been done is made using quantities and features inherent to the data set. The lower the DBI index value, the better is the clustering. It also has a drawback. A good value reported by this method does not imply the best information retrieval. The DBI index for *k* number of clusters is defined as Equation (Equation 3):
(3)$$DB\left(u\right)=\frac{1}{k}\sum_{i=1}^{k}max_{i\neq j}\left\{\frac{\Delta \left(X_{i}\right)+\Delta \left(X_{j}\right)}{\delta (X_{i},X_{j})}\right\}$$Cophenetic correlation coefficient index (CPCC) measures the correlation between respective cophenetic matrix $P_{c}$ and the proximity matrix *P* of *X*. Because both of the matrices are symmetric and have their diagonal elements equal to 0, we only consider the $M=\frac{N(N-1)}{2}$ upper diagonal elements of $P_{c}$ and *P*. Let $d_{ij}$ and $c_{ij}$ be the (i,j) element of *P* and $P_{c}$, respectively. Subsequently, the CPCC index is defined as Equation (Equation 4).
(4)$$CPCC=\frac{\frac{1}{M}\sum_{i=1}^{N-1}\sum_{j=i+1}^{N}d_{ij}\cdot c_{ij}-\mu_{p}\cdot \mu_{c}}{\sqrt{\left(\left(\frac{1}{M}\right)\sum_{i=1}^{N-1}\sum_{j=i+1}^{N}d_{ij}^{2}-\mu_{p}^{2}\right)\left(\left(\frac{1}{M}\right)\sum_{i=1}^{N-1}\sum_{j=i+1}^{N}c_{ij}^{2}-\mu_{c}^{2}\right)}}$$$\mu_{p}=\frac{2}{N(N-1)}\sum_{i=1}^{N}\sum_{j=i+1}^{N}d_{ij}$ and $\mu_{c}=\frac{2}{N(N-1)}\sum_{i=1}^{N}\sum_{j=i+1}^{N}c_{ij}$. The values of the CPCC are between −1 and +1. The closer the CPCC index to 1, the better the agreement between the cophenetic and the proximity matrix.Sillhouette index measures how well an observation is clustered and it estimates the average distance between clusters. For data point i∈C let $a\left(i\right)=\frac{1}{|C_{i}|-1}\sum_{j\in C,j\neq i}^{}d\left(i,j\right)$ be the mean distance between *i* and all other data points in the same cluster, where d(i,j) is the distance between data points *i* and *j* in the cluster $C_{i}$. We can interpret a(i) as a measure of how well *i* is assigned to its cluster (the smaller the value, the better the assignment). Subsequently, we define the mean dissimilarity of point *i* to some cluster *C* as the mean of the distance from *i* to all points in *C* (where $C\neq C_{i}$). For each data point $i\in C_{i}$, we now define $b\left(i\right)=min_{k\neq i}\frac{1}{|C_{k}|}\sum_{j\in C_{k}}d\left(i,j\right)$ to be the smallest (hence, the min operator in the formula) mean distance of *i* to all points in any other cluster, of which *i* is not a member. The cluster with this smallest mean dissimilarity is said to be the neighboring cluster of *i*, because it is the next best fit cluster for point *i*. We now define a Silhouette value of one data point *i* (see Equation (Equation 5)):
(5)$$s\left(i\right)=\frac{b(i)-a(i)}{max\{a(i),b(i)\}}\;\mathrm{if}\;|C_{i}\left|>1\;\mathrm{and}\;s\left(i\right)=0\;\mathrm{if}\;\right|C_{i}|=1.$$From the above definition it is clear that −1≤s(i)≤1. Also, note that score is 0 for clusters with size equal to 1.

The three best global criteria for assessing the quality of clustering are as follows:Calinski & Harabasz (Pseudo F) index (Equation (Equation 6)):
(6)$$G1\left(u\right)=\frac{tr(B)\backslash (u-1)}{tr(W)\backslash (n-u)}$$
where *B* is the inter-class covariance matrix, *W*—the inter-class covariance matrix, tr—the matrix trace, *u*—is the number of classes and *n* is the number of objects to be clustered.Hubert & Levine index (Equation (Equation 7)):
(7)$$G2\left(u\right)=\frac{D\left(u\right)-r\cdot D_{min}}{r\cdot D_{max}-r\cdot D_{min}}$$
where D(u) is the matrix of all inter-class distances, *r*—number of inter-class distances, $D_{min}$—smallest inter-class distance, $D_{max}$—longest inter-class distance.Gamma Baker & Hubert (Gamma) index (Equation (Equation 8)):
(8)$$G3\left(u\right)=\frac{s(+)-s(-)}{s(+)+s(-)}$$
where s(+) is the number of compatible distance pairs and s(−) is the number of incompatible distance pairs. If the intra-class distance is smaller than the inter-class distance then we consider the pair as compatible. Such comparisons are r·c, where *r* is the number of inter-class comparisons and *c* is the number of intra-class comparisons.

In this research, the authors use seven indices: Silhouette, Dunn, DBI, Calinski & Harabasz (Pseudo F), Hubert & Levine index, Gamma index, and CPCC.

## 3. Outliers in Rules

Outlier observations often appear in real data sets and require special attention, as they can have a significant impact on the analysis results. In non-hierarchical algorithms, outliers affect the quality of the created groups and their representatives, while, in hierarchical algorithms, outliers are most often small groups (one-element). The presence of unusual elements makes it difficult to properly use the knowledge that is contained in the data, and their elimination or improvement enables their more effective analysis. Rules are a common method of expressing the logic of the operation of information systems. Each rule-based system consists of a KB and an application mechanism that allows for the interpretation of these rules. Outstanding values in rules mean unprecedented rules that are rare as compared to others and they should be more thoroughly examined and analyzed by experts in the field in question, with a view to extending them as far as possible in the future the KB in an area not yet explored. Unusual elements that are not the result of errors often completely set out new trends or unexpected dependencies and their information content may be of significant positive importance for the formulation of valuable research hypotheses. We reduce the quality of the clusters and their representatives by clustering together rules that are not sufficiently similar, which decreases the inference efficiency.

### 3.1. Definition of an Outlier

Outlier detection is an important branch in the area of data mining. It is concerned with discovering the exceptional behaviors of certain objects. Outlier detection is a process of detecting data points in which their patterns deviate significantly from others. This definition may vary slightly based on the area that it is used. Hawkins (Hawkins, 1980) defines an outlier as an observation that deviates so much from other observations as to arouse suspicion that it was generated by a different mechanism. Outliers in the context of the rule-based KBs will be rules that do not match the rest of the knowledge (rules) stored in the KB.

### 3.2. Outliers Detection Algorithms and a Brief Review of the Literature

Detecting outliers that are grossly different from or inconsistent with the remaining data set is a major challenge in real-world knowledge discovery (mining) in database applications. An unusual observation is an observation that is inconsistent with other observations in the data set. Outliers often contain useful information regarding the system’s abnormal state that affects the data generation process. Methods that are based on data distribution, data distance, density, and grouping are among the most common methods for detecting observations that differ from the others. A number of studies on outliers detection have already been carried out, and several examples are listed below, depending on the method used. Some representative methods include algorithms:density-based: LOF (local outlier factor) [8], COF (connectivity-based outlier factor) [9],distance-based: LDOF (local distance-based outlier factor) [10], andclustering-based: TONMF (text outliers using non-negative matrix factorization) [11], OFP (outlier finding process) [12], FindOut [13], FindCBLOF (cluster-based local outlier factor) [14], CBOD (clustering-based outlier detection) [15], PLDOF [16], algorithm FCM (fuzzy c-means) [17], *K*-SVM [18], and ONION framework [19].

There are also deep learning based outlier detection methods, such as deep one-class SVM (support vector machine) [20] and GAN-based (generative adversarial network) methods [21,22,23].

### 3.3. Algorithms LOF and COF

The density-based local outlier factor (LOF) and connectivity-based outlier factor (COF) methods are both effective and simple than other techniques. LOF was proposed in 2000 by Markus M. Breuning, Hans-Peter Kriegel, Raymond T. Nga, and Jörg Sander [8], and is the most commonly used local method. The local outlier factor is based on a concept of a local density, where locality is given by *k* nearest neighbors, whose distance is used to estimate the density. The advantage of LOF is its ease of interpretation and ability to capture outliers that cannot be detected by a global approach. Several variants of LOF were also developed, including the outlier based on COF communication that was proposed in 2002 by Jian Tang, Zhixiang Chen, Ada Waichee Fu, and David W. Cheung [9]. COF is a variation of LOF, the difference is that the *k*-distance of object *p* is calculated in incremental mode. This algorithm was introduced as a result of the following observation: although a high-density set may represent a pattern, not all of the patterns need to be of high density. Unlike density-based methods, COF can detect exceptions among data of different densities. Additionally, the COF algorithm allows for the detection of exceptions that are close to high-density areas. COF indicates how far from the pattern the object under consideration is. The COF is similar to the LOF, but the density estimation for records is performed in different ways. In LOF, the *k*-nearest-neighbors (*k*-NN) are selected based on Euclidean distance. This indirectly assumes, that the data is distributed in a spherical way around the instance. If this assumptions is violated, for example if features have a direct linear correlation, the density estimation is incorrect. COF wants to compensate this shortcoming and estimates the local density for the neighborhood while using a shortest-path approach, called the chaining distance. Both COF and LOF algorithms are dedicated to use with numerical data. What is more, they were not examined in the context of complex data, like rules *If-Then* and rule clusters.

#### LOF (Local Outlier Factor) Example

Let consider the following 10 data points: A(1,1), B(1.5,1.5), C(1.5,2), D(5,8), E(5,6), F(6,7), G(7,3.5), H(8,4), I(8,3.5), and $J^{\prime }(8.5,1)$. LOF algorithm runs in 6 steps. The main goal of the algorithm is to calculate the LOF value for each point and show the top one outlier for a given number of cluster *k* and a given distance measure. Let set a *k* to value 2 and distance measure as Euclidean distance. Figure 1 presents the visualization of data points and two values LOF and COF for each data point.

In the first step of the LOF algorithm it is necessary for calculating the distance between each pair of data point (see Table 1).

The second step is based on calculating, for all of the data points, the distance between a given data point *o* and its *k*-th nearest neighbor. It requires to search Table 1 and for each data point *o* to return its distance to the second nearest neighbor. For the data point *A* the nearest neighbors are *B* and *C*, but the second nearest neighbor (2−NN) is *C* that is why the $dist_{2}\left(A\right)=dist\left(A,C\right)=1.1180$. For the rest of data points, the $dist_{2}\left(o\right)$ are as follows: $dist_{2}\left(B\right)=dist\left(A,B\right)=0.7071$ because *A* is the 2−NN, $dist_{2}\left(C\right)=dist\left(A,C\right)=1.1180$, $dist_{2}\left(D\right)=dist\left(D,E\right)=2$, $dist_{2}\left(E\right)=dist\left(D,E\right)=2$, $dist_{2}\left(F\right)=dist\left(E,F\right)=1.4142$, $dist_{2}\left(G\right)=dist\left(G,H\right)=1.1180$, $dist_{2}\left(H\right)=dist\left(G,H\right)=1.1180$, $dist_{2}\left(I\right)=dist\left(G,I\right)=1$ and $dist_{2}\left(J^{\prime }\right)=dist\left(G,J^{\prime }\right)=2.9154$ because *G* is the 2−NN of $J^{\prime }$. In step 3, for each data point *o* the *k*-distance neighborhood of *o* is returned as $N_{k}\left(o\right)=\left\{o^{\prime }∥o^{\prime }\in D,dost\left(o,o^{\prime }\right)\leq dist\left(o\right)\right\}$. Therefore, *k*-distance neighborhood of *A* ($N_{2}\left(A\right)$) is {B,C} because only for *B* and *C* their distance to *A* is less or equal to $dist_{2}\left(A\right)$. For the rest of data points $N_{2}\left(o\right)$ is following: $N_{2}\left(B\right)=\left\{A,C\right\}$, $N_{2}\left(C\right)=\left\{B,A\right\}$, $N_{2}\left(D\right)=\left\{E,F\right\}$, $N_{2}\left(E\right)=\left\{D,F\right\}$, $N_{2}\left(F\right)=\left\{D,E\right\}$, $N_{2}\left(G\right)=\left\{H,I\right\}$, $N_{2}\left(H\right)=\left\{G,I\right\}$, $N_{2}\left(I\right)=\left\{G,H\right\}$ and $N_{2}\left(J^{\prime }\right)=\left\{G,I\right\}$.

Step 4 is based on calculating for each data point *o* the so called local reachability density $lrd_{k}\left(o\right)$, according to Equation (Equation 9).
(9)$$lrd_{k}\left(o\right)=\frac{\left|\right|N_{k}\left(o\right)\left|\right|}{\sum_{o^{\prime }\in N_{k}\left(o\right)}reach-dist_{k}\left(o^{\prime }\leftarrow o\right)}$$
where $\left|\right|N_{k}\left(o\right)\left|\right|$ means the number of objects in $N_{k}\left(o\right)$, for example: $\left|\right|N_{2}\left(A\right)\left|\right|=\left\{B,C\right\}=2$ and $reach-dist_{k}\left(o^{\prime }\leftarrow o\right)=max\left\{dist_{k}\left(o\right),dist\left(o,o^{\prime }\right)\right\}$. Thus, $lrd_{k}\left(o\right)$ for k=2 and data point $o\in \{A,B,\ldots ,J^{\prime }\}$ are as follows: $lrd_{2}\left(A\right)=1.0958$, $lrd_{2}\left(B\right)=0.8944$, $lrd_{2}\left(C\right)=1.0958$, $lrd_{2}\left(D\right)=0.5857$, $lrd_{2}\left(E\right)=0.5857$, $lrd_{2}\left(F\right)=0.5$, $lrd_{2}\left(G\right)=0.9442$, $lrd_{2}\left(H\right)=0.9442$, $lrd_{2}\left(I\right)=0.8944$, $lrd_{2}\left(J^{\prime }\right)=0.3659$. The explanation fo calculating $lrd_{2}\left(A\right)$ is as follows: $lrd_{2}\left(A\right)=\frac{\left(||N_{2}\left(A\right)||\right)}{\left(reach-dist_{2}\left(B\leftarrow A\right)+reach-dist_{2}\left(C\leftarrow A\right)\right)}$ where $reach-dist_{2}\left(B\leftarrow A\right)=max\left\{dist_{2}\left(B\right),dist\left(B,A\right)\right\}=max\left\{0.7071,0.7071\right\}=0.7071$ and $reach-dist_{2}\left(C\leftarrow A\right)=max\left\{dist_{2}\left(C\right),dist\left(C,A\right)\right\}=max\left\{1.1180,1.1180\right\}=1.1180$. Thus, $lrd_{2}\left(A\right)=\frac{2}{\left(0.7071+1.1180\right)}=1.0958$.

Step 5 requires calculating for each data point *o* the final LOF value according to Equation (Equation 10). The LOF of a point *o* is the sum of the $lrd_{k}\left(o\right)$ of all the points in the set $n_{k}\left(o\right)$ multiplied by the sum of the $reach-dist_{k}\left(o\right)$ of all the points of the same set, to the point *o*, all divided by the number of items in the set, ||N−k(o)||, squared.
(10)$$LOF_{k}\left(o\right)=\sum_{o^{\prime }\in N_{k}\left(o\right)}lrd_{k}\left(o^{\prime }\right){\dot{\sum }}_{o^{\prime }\in N_{k}\left(o\right)}reach-dist_{k}\left(o^{\prime }\leftarrow o\right)$$For data point *A* this process runs, as follows: $LOF_{2}\left(A\right)=\frac{\left(lrd_{2}\left(B\right)+lrd_{2}\left(C\right)\right)\dot{(}reach-dist_{2}\left(B\leftarrow A\right)+reach-dist_{2}\left(C\leftarrow A\right))}{\left|\right|N_{2}{\left(A\right)}^{2}\left|\right|}$
$=\frac{\left(\left(0.8944+1.0958\right)\dot{(}0.7071+1.1180)\right)}{\left(2*2\right)}=\frac{3.6324}{4}=0.9081$. For other data points $B,\ldots ,J^{\prime }$, the LOF values are following: $LOF_{2}\left(J^{\prime }\right)=2.5121$, $LOF_{2}\left(B\right)=1.2251$, $LOF_{2}\left(I\right)=1.0557$, $LOF_{2}\left(G\right)=0.9736$, $LOF_{2}\left(H\right)=0.9736$, $LOF_{2}\left(D\right)=0.9267$, $LOF_{2}\left(E\right)=0.9267$ and $LOF_{2}\left(C\right)=0.9081$.

In the last step, it is necessary to sort LOF values and choose the higher $LOF_{k}\left(o\right)$ value, which here is $LOF_{2}\left(J^{\prime }\right)=2.5152$. This means that data point $J^{\prime }$ is the outlier.

In case of using COF algorithm, six steps are also executed. The first step is based on finding *k*-NN (for k=1) of the data point *o*. For each data *o*, it is necessary to find set $N_{k}\left(o\right)$ of its *k*-NN. This process is also based on the Table 1. For example, for *A* the nearest neighbor is *B*. Subsequently, in step 2, it is necessary to find Set based nearest (SBN) path which represents *k* nearest data points in order s={o1,o2,…,ok}. For data point *A*
SBNpath={A,B}, as it arranges data points in such a way that it should create a path. In step 3, we need to find set based nearest (SBN) trail that represents a sequence of edges based on SBN path $e=\{e_{1},e_{2},\ldots ,e_{k}\}$. For *A*
SBNtrail={(A,B)} as it arranges set of data points with respect to edges $e_{1}$ respectively. Step 4 is responsible for finding the cost of SBN trail which represents the distance between two data point (edge value). The SBN trail is the sequence of edges $e=\{e_{1},\ldots ,e_{k}\}$, where each edge connects two consecutive nearest neighbours from the SBN path. For point *A* Cost description = {0.7071} weight of each edge (in this case only $e_{1}$). The goal of next step, 5, is to find Average chaining distance $ac-dist_{N_{k}\left(o\right)\cup o}\left(o\right)$ of the data point and finally, in step 6 it is possible to calculate connectivity-based outlier factor COF value of the data point *o* with respect to its *k*-neighbourhood. COF is the ratio of average chaining distance of data point and the average of average chaining distance of *k*-NN of the data point.

### 3.4. Algorithm K-MEANS

We can also detect outliers by clustering using the *K*-MEANS algorithm. Using *K*-MEANS, data are divided into *k* groups, assigning them to the nearest cluster centres. Subsequently, we can calculate the distance (or dissimilarity) between each object and its cluster center and select those with the greatest distances as outliers. *K*-MEANS clustering is one of the simplest and popular unsupervised machine learning algorithms. Traditional *K*-MEANS algorithm consists of the following steps:choose *k* initial centers,assign data points *X* to the closest center,recompute and find new centers, andrepeat step 2 and 3 until the centers no longer change. Center of groups of the *k*-clusters can be used to detect outliers. After building the clusters, we mark points that are far from the centers of the *k*-clusters. In other words, we consider points far from the center of the cluster to which they belong (in terms of distance) as outliers.For each object:
select the cluster to which the given object belongs andcalculate the distance between each object and the center of the cluster to which it belongs.For each cluster, select the objects with the greatest distance (in the sense of the Euclidean measure) as outlier.

Assuming that every data set contains the feature DECISION_VALUES, we may take as *k* a number that is equal to DECISION_VALUES (number of decision values). The algorithm performs very well when the data set elements form a natural collection with different characteristics (properties), the more diverse the data sets, the less iterations needed to detect the internal structure in the data set. The undoubted advantages of the *k*-means algorithm are its simplicity and linear time complexity in relation to the number of objects, which makes it attractive for processing large data sets [24]. It also has a few disadvantages:is very sensitive to noisy data or data containing peculiar points, as such it points significantly affect average clusters causing their distortion,the result of the algorithm, i.e., the final division of objects between clusters, strongly depends on the initial division of the objects and depending on the initial choice of the center of the cluster, it can generate different sets of clusters,you should know the number of clusters in advance, andonly suitable for numerical data analysis.

In conclusion, typical observation is concentrated with others, while unusual observations are not part of any group. Looking at the *K*-MEANS algorithm, you can see that typical observations are closer to the group’s centre of gravity, while unusual observations are far from the nearest centre of gravity.

### 3.5. Algorithm SMALL CLUSTER

The SMALL
CLUSTERS algorithm was proposed and implemented by the authors. It uses the hierarchical clustering method. The detection of outliers is a component of clustering algorithms, objects that cannot be included in any cluster can be treated as outliers. We can also treat rules in a small groups (which could not be combined with larger groups, having unusual or rare attribute values) as unusual. The applied methodology proposes using the size of the resulting clusters as indicators of the presence of outliers. It assumes that outlier observations will be distant in terms of the metric used for clustering from normal and frequent observations and will, therefore, be isolated into smaller groups. The pseudocode of the SMALL
CLUSTERS algorithm to identify outliers in a data set is presented as Algorithm 1.
**Algorithm 1** Algorithm SMALL
CLUSTERS**Require:**DATA—a data set containing *k* variables and *n* items; dist—distance function; AHC—agglomerative hierarchical clustering algorithm; nc—the number of clusters to be used (this number implies the level of the tree-dendrogram cut); and, *t*—threshold determining the number of small clusters.**Ensure:**out put—a set of outlier observations. Obtain a distance *D* matrix using distance dist to the selected data set Use the AHC algorithm to build a dendrogram using the distance matrix *D* Cut the dendrogram at *l* level, which determines how many clusters (nc number) we want to cut the tree For each cluster *C* created after cut the dendrogram, where *i* is the number of the next group formed after cut the dendrogram, i∈N, i≥2, do **if** number of observations in the cluster $C_{i}<t$
**then**  output ← output ∪{observations $\in C_{i}$} **end if**


Figure 2 presents a dendrogram with four clusters before removing unusual rules using SMALL
CLUSTERS algorithm in which the small one-element group in red responds to potential outlier.

## 4. Application Rules Analysis

The web application in the form of a research cockpit was developed as a result of an analysis of the currently available solutions. Many known algorithms for clustering and outliers detection have already been implemented in the *R* or Python environment. Nevertheless, it was required software, which, for the needs of the conducted analyses, would provide specific functionalities allowing to verify the assumed thesis of the work. The analyses showed that there is no similar tool for analyzing rule-based KBs and identifying outliers, and there is no tool that examines the consistency and quality of rule groups. The implemented software in the form of a research cockpit allows for the analysis of a data set of various sizes and data types. The analysis includes the separation of unusual rules, followed by a hierarchical clustering of rules (without outlier rules) and descriptive characteristics for the clustering quality indicators used, as well as graphic characteristics of the obtained clusters. The software provides ready-made methods for transforming various types of data. The authors analyze complex data, i.e., domain-specific databases of knowledge about rule-based representation. To start exploring, the application had to ensure that the relevant data can be extracted from a file containing decision-making rules and converted to a suitable format—data frame. In addition, the research cockpit located at: https://studies.shinyapps.io/RulesAnalysis allows for you to analyze a set of data of different sizes and types. It maximally simplifies the path from downloading data, by removing unusual rules while using the LOF, COF, *K*-MEANS, and SMALL
CLUSTERS algorithms, clustering and quality assessment (Figure 3), for the visualization of clusters (Figure 4) with the final table summarizing the analysis, while allowing for simple change of the parameters that are relevant to the research. It is possible to see that the rule 10 was classified as an outlier (Figure 3) and, in Figure 4, we may observe the dendrogram with and without this rule. Such a solution was not found among the available *R* packages or modules in Python.

## 5. Research Work

The major disadvantage of most of the clustering algorithms is their sensitivity to the presence of outliers, which strongly affects the deterioration of the resulting groups. An effective way of tackling this phenomenon is to isolate outliers in the initial stage and, thus, prepare data for clustering. The authors assume that the removed outliers will be passed on to the expert in the knowledge field for analysis as rare and nontrivial rules.

### 5.1. Materials and Methods

Algorithms for detecting outliers (unusual rules) in the rule-based KBs: LOF, COF, *K*-MEANS, and SMALL
CLUSTERS, were implemented and subjected to experiments using the presented research cockpit.

In the first step, the rule-based KB is loaded in the form of an RSES [4] file (this is a file with the extension of rul), which contains rules saved in the order they were generated. This is a flat structure of the rule-based KB (the rules are written as Horn clauses). The KB in RSES format is written, as follows [25]:
the first line is the name of the loaded KB with rules,the second line specifies the number of *z* conditional and *m* decision attributes,the next z+m lines are the names and types of conditional and decision attributes. The number at the end of the line describing the conditional attributes specifies the length of the maximum attribute value, counted in printed characters,the next line specifies the number *u* of the value of the decision attribute. Traditionally, the decision attribute is entered last in the order in which it appears in the rule, but this is not required,the next *u* lines determine the values of the decision attribute, andin the last section of the file, the header specifying the number of *n* rules in the KB is followed by *n* lines with the rules of the KB. The last value in each line determines the so-called rule support, i.e., the number of cases in the original database confirming it.The next step is rule clustering, which is carried out thanks to a AHC algorithm that analyses the similarity of rules. The data must be in the form of a matrix (data frame), whose rows represent each rule and the columns represent its successive characteristics.The web application created for research purposes transforms the original KB with rules containing a different number of premises into a matrix, in which the lines match the rules and the columns contain all possible features in the premises of the rules with an attribute condition. This matrix allows for you to specify the similarity between rules and combine similar rules into groups. Initially, each rule is treated as a separate (one-element) cluster, based on the Gower distance measure, sequential clustering (i.e., agglomeration) of rules is performed, depending on how different or similar they are. The first cluster is always created from the combination of the two most similar rules, in subsequent stages, either a single rule is added to an existing cluster or the clusters are combined, also according to the shortest distance. As a result of repetition of the procedure—combining the two closest clusters, calculating the distance, etc.—we obtain one final group of rules and a hierarchical order chart. Such a graph, called a dendrogram, shows the essential features of the merging performed, externalizes the order of merging rules into clusters and the levels at which the rules were merged for the first time. At the end, the appropriate function cuts the resulting dendrogram to the desired number of clusters (a parameter that can be interactively controlled in the program). The dendrogram is only a certain summary of the information that is contained in the distance matrix; while using the cophenetic correlation coefficient CPCC, we obtain information about the desired number of groups to which dendrogram should be cut. In the research, CPCC value was determined for each of the seven cluster linking methods (single, complete, average, centroid, mcquitty, ward.D, and ward.D2), in addition, the resulting dendrograms were also visually assessed and the optimal number of groups to which the created connection tree should be cut was selected.After the clustering proces, a quality analysis was performed taking into account the seven indicators Silhouette, Dunn, DBI, Calinski & Harabasz (Pseudo F), Hubert & Levine, Gamma, and CPCC. In our research, we want to learn about unusual rules, which constitute 1%, 5%, 10% of all rules in the KB using LOF, COF, *K*-MEANS methods, whereas, in the case of the SMALL
CLUSTERS method, it was assumed that unusual rules are rules that will create small groups containing no more than 5% of rules from the analyzed KBs.After detecting unusual rules using the above-mentioned algorithms and then removing them from the KB (unusual rules are passed on to a field expert for analysis), the process of clustering was conducted again. According to the thesis of the work, the quality of created clusters should be improved.In order to verify this, all of the clustering quality indicators were recalculated and the number of cases in which the quality of clusters has improved was verified. We also wanted to know how many indicators showed a deterioration in the quality of newly formed groups, and how many remained unchanged and did not react to the removal of unusual rules. It turns out that sometimes removing even a large number of outliers does not improve cluster quality. After this step, the results were analyzed and selected indicators were evaluated for consistency of results.

### 5.2. Data Source Description

The source for creating rule-based KBs was primarily the Machine Learning repository, the Kaggle repository, and the GitHub service and the collections of different sizes and structures of stored data contained therein, in addition, the krukenberg database was used, which is a KB created for the medical domain. The authors focus on the representation of knowledge in the form of rules generated automatically from data using the rough set theory, using the Learning from Examples Module algorithm (LEM2) proposed in [26] and implemented in the RSES system. The idea of rough set was proposed by Pawlak (1982) [27] as a new mathematical tool to deal with vague concepts (Rough Set theory is an effective approach for data analysis and its main goal is synthesizing approximation of a crisp set in terms of a pair of sets which give the upper and lower approximation of the original set). Rule induction is a part of machine learning in which the rules are extracted from set of observations. Rules generation always has an important role in data mining and provides some connection between attributes that are helpful for decision making. The process for extracting rules from the original data set is as follows. Databases have been loaded into RSES program in which using LEM2 [28] algorithm, minimal rules were induced as input rules database. All of the experiments presented in this chapter were performed on these rule bases. The LEM2 algorithm allows for you to search for complex rules based on decision class approximation. Depending on the type of approximation, rules are generated:certain—based on lower approximation,possible—based on upper approximation, andrough—based on the boundary region of approximation.

The algorithm looks for a condition that would be met by as many objects belonging to the decision class as possible. If more than one condition (with the same support) is present, then the one with the least support by all of the objects is selected. The selected condition creates the beginning of the rule, which is extended with further conditions in a similar way. The generated rule is then trimmed, i.e., the conditions are removed, the absence of which does not result in the set of conditions covering objects outside the approximation. The next rule is only built on the basis of objects that were adopted at the beginning of the approximation that have not been covered by rules that have been built up to now. When there are no more such objects, the generation of rules for a given decision class ends and unnecessary rules are removed, i.e., those whose absence does not affect the coverage of all objects of a given approximation by the set of rules.

The authors used six different KBs in their research:the weather database, also known as tennis, is a simple set of data with decisions about playing tennis depending on the weather. The database consists of only 14 records. Each record contains four weather information that can be useful for making decisions, the last column represents the decision class. Here we have qualitative and logical attributes.The independent variables are Outlook, Temperature, Humidity, and Wind. The dependent variable is whether to play tennis or not,the diabetes database (Pima Indians Diabetes Database)—this data set is originally from the National Institute of Diabetes and Digestive and Kidney Diseases. The objective of the data set is to diagnostically predict whether or not a patient has diabetes, based on certain diagnostic measurements included in the data set. Several constraints were placed on the selection of these instances from a larger database. In particular, all patients here are females at least 21 years old of Pima Indian heritage. The data set is 768 observations, each record consists of eight medical predictive variables and one Outcome target variable, these are quantitative data. The last column of the data set indicates if the person has been diagnosed with diabetes (1) or not (0). Predictor variables includes the number of pregnancies the patient has had, their BMI, insulin level, age, and so on,the heart disease database (the authors of the databases have requested that any publications resulting from the use of the data include the names of the principal investigator responsible for the data collection at each institution. They would be: 1. Hungarian Institute of Cardiology. Budapest: Andras Janosi, M.D. 2. University Hospital, Zurich, Switzerland: William Steinbrunn, M.D. 3. University Hospital, Basel, Switzerland: Matthias Pfisterer, M.D. 4. V.A. Medical Center, Long Beach and Cleveland Clinic Foundation:Robert Detrano, M.D., Ph.D.) is a data set containing 303 observations from 1988, consisting of four databases: Cleveland, Hungary, Switzerland and long Beach. This database contains 76 attributes (including the decision attribute), but all published experiments refer to using a subset of 14 of them. The data set consists of 13 numeric attributes, including age, sex, chest pain type, resting blood pressure, cholesterol, fasting blood sugar, resting ECG, maximum heart rate, exercise induced angina, oldpeak, slope, number of vessels colored, and thal. The classes comprise of integers valued 0 (no presence of heart disease) and 1 (presence of heart disease).the krukenberg database, which is a KB created for the medical domain thematically related to the krukenberg tumor. A Krukenberg tumor refers to a malignancy in the ovary that metastasized from a primary site, classically the gastrointestinal tract, although it can arise in other tissues such as the breast. This is a set of 111 observations, each with 23 qualitative attributes (among others, age, sex, weight, number of lymph nodes removed, etc.), which contains a decision attribute that can take nine different values and determines the patient’s condition,nursery database is a database of nursery schools. Database was derived from a hierarchical decision model originally developed to rank applications for nursery schools. It was used during several years in 1980’s when there was excessive enrollment to these schools in Ljubljana, Slovenia, and the rejected applications frequently needed an objective explanation. The final decision depended on three subproblems: occupation of parents and child’s nursery, family structure and financial standing, and social and health picture of the family. The model was developed within expert system shell for decision making DEX. The set contains 12,960 instances, with each instance consisting of nine quality attributes, including the decision class with five values: (recommend, priority, not_recom, very_recom, spec_prior), andlibra database (Libras Movement), is a set from the University of São Paulo in Brazil, data set contains 15 classes of 24 instances each (a total of 360 instances), where each class references to a hand movement type in LIBRAS (Brazilian Sign Language (BSL) is the sign language used by deaf communities of urban Brazil). In the video pre-processing, a time normalization is carried out selecting 45 frames from each video, in accordance to a uniform distribution. In each frame, the centroid pixels of the segmented objects (the hand) are found, which compose the discrete version of the curve F with 45 points. All of the curves are normalized in the unitary space. In order to prepare these movements to be analyzed by algorithms, we have carried out a mapping operation, which is, each curve F is mapped in a representation with 90 features, representing the coordinates of movement. The set contains 90 numeric (double) attributes and one decision attribute for the class (integer).Each time that we get rules from the original data set. For example, the diabetes data set contains 768 observations described with eight continuous attributes, the objects are divided into two decision classes (268 of them are people who have been positively tested for diabetes—1, while 500 observations represent people who have not been diagnosed with diabetes—0). When these data were processed, system RSES developed 490 rules using the LEM2 algorithm. Most often, when the size of input data increases, the number of generated rules also increases.

A brief description of each KB is shown in Table 2. The bases created differ in the number of rules, the number of attributes, the type of attributes and the length of rules. The data sets are available at https://github.com/1662lacsap/RulesAnalysis/blob/master/Data%20sets%20in%20experiments.zip.

The experiments can be summarised in four points.

The following experiments were conducted on each of the six KBs:
(a)For each of the seven measures of distance between clusters (single, complete, average, centroid, mcquitty, ward.D and ward.D2), the CPCC coefficient was determined and visual evaluation of the dendrogram was performed. In this way, information on the best method of grouping was obtained, which will allow for obtaining the optimal number of clusters for a given KB.(b)After selecting the best distance measure, the optimum number of clusters *k* was selected (k=2,3,…, increasing *k* by 1 in each step up to 10) and each time all 7 clustering quality indicators were calculated, all of the indicators were compared with their values for other *k* values each time and a visual evaluation of the dendrogram was re-conducted—after such an assessment, one best (optimal) number of clusters was selected for which a given KB will be divided.After these two steps ((a) and (b)), the best method of distance between the clusters and the optimal number of clusters were found and only these were taken into account in the further analysis. The values of the calculated indicators were remembered for comparison at a later stage of the research.For each KB, using the LOF, COF, *K*-MEANS methods, unusual rules were found, which are 1%, 5%, 10% of all rules in the KB, whereas, in the case of the SMALL
CLUSTERS method, it was assumed that unusual rules are rules that will form small groups that contain no more than 5% of rules from the analyzed KB. For LOF and COF methods *k*-distance was selected in the range from 1 to 150, for *K*-MEANS method seed parameter was controlled in the range from 1 to 200, and for the SMALL
CLUSTERS method, the nc parameter of the algorithm was set appropriately in the range from 2 to 12 in order to divide the examined set into groups and extract small clusters. By controlling these parameters for each algorithm, it was sought to find the best (optimal) solution for which all of the indicators will show an improvement in clustering quality after removing the outliers (unusual rules) found by these algorithms.After finding and removing the outliers from the KB, the values of the recalculated clustering quality indicators were remembered, and then the results of the research were analyzed and selected indicators were assessed for the consistency of results.

Taking into account the smallest KB (*weather*) with five rules it is quite easy to see which rule (and why) is assigned as an outlier. The KB contains five rules:(humidity=normal)&(windy=FALSE)=>(play=yes)(outlook=overcast)=>(play=yes)(humidity=high)&(outlook=sunny)=>(play=no)(outlook=rainy)&(windy=TRUE)=>(play=no)(temperature=mild)&(outlook=rainy)&(humidity=high)&(windy=FALSE)=>(play=yes)

According to the results, using the LOF and COF algorithm rule 1: (humidity=normal)&(windy=FALSE)=>(play=yes) was assigned as an outlier, while, according to the *K*-MEANS algorithm rule 3: (humidity=high)&(outlook=sunny)=>(play=no) was classified as an outlier. In both cases, these rules were clearly different from the rest of the rules. Either they had different attributes in the description or a different length (number of attributes). In the case of SMALL
CLUSTERS algorithm, none of the rules were classified as an outlier. The reason is that this KB is too small to contain many clusters with many rules. That is why it was not able to find it in a small rule cluster that would have smaller size when comparing to others.

When we take greather KB, for example, heart disease, with 99 rules, the results of discovering 10% of outliers (for 99 of rules it is 10 rules) are presented in Table 3. Rows only present those rules that were at least once classified as outliers. The + sign in a given row and column means that the rule with the index given in the first column was marked as an outlier by the method from the indicated column. It can be seen that such rules numbered by 10 and 11 were assigned as outliers for each of the analyzed outlier detection metod: LOF, COF, *K*-MEANS, SMALL
CLUSTERS. These two rules looks totally different (use different attributes and values of these attributes) comparing to the other 97 of rules.

## 6. Results

The purpose of using clustering quality indicators is to seek an answer to the question: to what extent does the obtained group structure resulting from a given clustering method represent a good summary of the information contained in the data? The tables and charts cover selected aspects of the authors’ research.

### 6.1. Cluster Quality Measurement vs. Methods for Detecting Unusual Rules and Their Influence on the Frequency of Improvement or Deterioration of Cluster Quality

After the detection of unusual rules using LOF, COF, *K*-MEANS, SMALL
CLUSTERS algorithms, and then removing them from the KB, the clustering process was carried out again and, according to the thesis of the work, the quality of created clusters should improve. In order to check this, all of the grouping quality indicators were recalculated and it was verified in how many cases the quality of clusters improved, how many indicators showed a deterioration in the quality of newly created groups, and how many remained unchanged and did not react to the removal of unusual rules.

Figure 5 shows the result of the analysis. It can be noted that, only in the case of three cluster quality measures, i.e., the DBI index, the Pseudo F measure, and the CPCC coefficient each time in the case of the LOF and COF methods, and in the case of the CPCC coefficient, also for the *K*-MEANS method, the quality of clusters is improved. This means that these indices are very sensitive to outliers in a given data set. When using the Silhouette index or the Hubert & Levine index, we sometimes observed a deterioration in the quality of the groups after removing unusual rules for all methods, this deterioration appeared from time-to-time also for other indices, especially for *K*-MEANS and SMALL
CLUSTERS methods.

The least improvement in the quality of clusters after removing the outliers was recorded by the Dunn index (in the case of the SMALL
CLUSTERS method only in less than 19% of cases). The second index that showed less improvement in the quality of clusters after removing unusual rules was the Hubert & Levine index—see Figure 6.

### 6.2. The Impact of the Number of Removed Outliers on Improving the Quality of Clusters

The authors also decided to check whether the improvement of the quality of the resulting groups depends on the number of removed outlier rules for selected quality indicators.

For this purpose, the cases where 1%, 5%, and 10% of unusual rules were removed from the KB under investigation were analyzed. In the case of the SMALL
CLUSTERS method, in which outliers are treated as small groups, clusters with no more than 5% unusual rules were removed. For data sets weather, libra, nursery, and heart disease it represents about 10%. In the case of the krukenberg set, it is 5% of the set and for diabetes set the removed rules were slightly more than 1% (1.63%) of the set.

Figure 7 shows the results.

There are indices that always show an improvement in the quality of the rule clusters when removing a small number of outliers (1%), such as the Silhouette index, the DBI index, and the CPCC. It would seem that removing more outliers would have a better effect, but the fewer outliers found and removed, the more often most indexes indicated an improvement in quality. Most often, the quality of the groups improved after removing 1% or possibly 5% outliers, except for the Dunn index, which shows the opposite tendency.

After a summary of the indications, it can be concluded that the removal of few of the most sensitive outliers is most effective in improving the quality of the clusters, and removing more and more of them (subsequent observations that are less and less outlier) is not so effective. Choosing the best clustering rating indicator is just as difficult as choosing the best clustering algorithm. The decision may vary, depending on the experiment being carried out, the type of data analyzed, and may sometimes be related to individual user preferences, e.g., a division containing strong group separation is more required than one that contains high internal consistency.

### 6.3. Comparison of Clustering Quality Indicators in Terms of the Frequency of Showing Improvement after Removing Outliers (Unusual Rules)

Table 4 shows which clustering quality indicators most often show improvement after removing outliers (unusual rules). We see that the Dunn index has the lowest effectiveness in showing improvement in cluster quality about 25%). This means that only one in four cases after removing the outliers of cluster quality according to this index has improved. We see an inconsistency here. As most indices more often define clusters as good, when analyzing the same sets after the division, from the rest of the indices, a little less, but the Hubert & Levine index also stands out of line. The best quality of clusters is shown by the CPCC (almost every time outliers are removed—close to 100%) and the DBI index, the Silhouette index, and the Pseudo F index are doing not much worse; these are the most sensitive indices to outliers appearing in a given KB.

### 6.4. Frequency of Improvement of Cluster Quality vs. Outliers Detection Method

The frequency of cluster quality improvement for the analyzed algorithms of outliers detection was also verified: LOF, COF, *K*-MEANS and SMALL
CLUSTERS—see Figure 8.

We can see that, most often, the improvement in the quality of clusters after removing anusual rules is achieved using two methods: LOF and COF. The difference is so small that both of the algorithms can be awarded first place ex aequo. Slightly worse (about 5% fewer cases), the SMALL
CLUSTERS algorithm has shown improvements in the quality of the groups. The *K*-MEANS algorithm proved to be the worst in all categories. The least frequently, it showed an improvement in the quality of clusters (72.22%). Most often, it showed a deterioration of the quality of clusters after removing unusual cases (11.91%), and most often did not react to the removed outliers (15.87%). However, these were not large differences in all of the examined algorithms, approximately the same extent (between 12–16%) they did not show that there was a change, when the outliers were removed, the COF algorithm showed deterioration the least frequently, but the difference to the worst in this category of *K*-MEANS algorithm is just over 5%.

### 6.5. Data Types and the Quality of Clustering

The last aspect of the research concerned the nature of the data. It was checked whether the type of data has an impact on the results achieved—see Figure 9. The detection of outliers in the rules allows for optimizing inferences in the rule-based KBs and, thus, increases the effectiveness of decision support systems. When comparing different KBs containing rules, we may notice that, for most of them, when custom rules are removed, metrics have shown improvements in the range of 74% to 86% of cases. The exception is a small set of weather, for which improvement was only achieved in about 62% of cases. It can be concluded that, for the majority of the analyzed sets, in at least 75% of cases, the indicators showed an improvement in the quality of the resulting clusters after removing unusual rules. This confirms the consistency of the indices used, and encourages the analysis of KBs in terms of outliers, as it is highly likely that we will improve their efficiency. It may be surprising that the libra and krukenberg KBs, which have most often improved, are data sets with more attributes than the others, unexpectedly the indicators hardly recorded any deterioration of groups in these databases. The size of the KB strongly influences the inference time; therefore, it seems necessary to modify the existing structure of the domain KB. Cluster analysis allows for you to group similar rules into clusters. However, the structure of the group can be easily destroyed by several outliers. The research has shown that the applied LOF, COF, *K*-MEANS, and SMALL
CLUSTERS methods are perfectly suitable for detecting outliers in the initial stage. The experiments on real data have confirmed the effectiveness of the approach. Unusual rules in the KB disturb the grouping result, and the quality of the groups obtained from the analysis depends significantly on the efficiency of the methods used to recognize outliers. The methods used combined with the agglomeration hierarchical algorithm have proven to be an effective tool for rule clustering in the KBs. The identification of unusual elements is important in many theoretical and application problems in various fields of science and practical activities. The early detection of unusual rules, which will then be passed on to field experts, will enrich the system with rare and valuable knowledge.

## 7. Discussion

To verify the impact of unusual rules on cluster quality, the authors analyzed many different outlier mining algorithms and finally chose four methods: LOF, COF, *K*-MEANS, and SMALL
CLUSTERS method proposed by them. Previously, these methods were used for typical simple data. In this work, they were used in order to investigate the effectiveness of detecting unusual (outlying) rules, as well as to verify the sensitivity of selected cluster quality measures to the presence of outlying values (unique rules) in KBs. Just like the problem of clustering data, the problem of cluster validity is very complex. Clustering may end with different divisions depending on the method selected and the parameters set. For this reason, there are a large number of indicators for assessing the quality of the resulting group structure. The different objectives set before clustering are imposed by different methods of assessing their implementation.There is no single universal indicator that can always be used, regardless of the problem being solved and the method used. Each of them takes into account only part of the information about the group structure in its construction. It is necessary to know a wide range of existing indicators and their basic properties in order to choose the right indicator for a given application. Therefore, the research used seven different clustering quality indicators to examine the quality of the resulting groups after removing a different number of unusual rules (1%, 5%, 10%) of the data. In the case of small clusters, it was assumed that a group containing up to 5% set is a so-called small group treated as an outliers. Six different KBs were used in the experimental phase. The research shows that the most optimal results were obtained for two algorithms in order to detect outliers of LOF and COF. The SMALL
CLUSTERS algorithm that was proposed by the authors showed an improvement in the quality of the groups slightly worse (about 5% less cases). The *K*-MEANS algorithm turned out to be the worst; however, it was not a big difference all of the algorithms studied, more or less to the same extent (between 12–16%) did not show any change, when the outliers were removed, the COF algorithm was the least likely to show deterioration, but the difference to the worst in this category of *K*-MEANS was only slightly over 5%. You can see that there are no major differences between the algorithms and that the quality indicators are consistent and show similar results.

It can be noticed that, in the case of the three cluster quality measures, i.e., the DBI index, the Pseudo F measure, and the CPCC coefficient each time in the case of the LOF and COF methods, and in the case of the CPCC also for the *K*-MEANS method, the quality of the clusters improved. This means that these indices are particularly sensitive to outliers in the data set. The Dunn index has the least improvement in cluster quality after removing outliers (in the case of SMALL
CLUSTERS method only in less than 19% of cases). The second index, which was less marked by an improvement in the quality of clusters after removing unusual rules, was the Hubert & Levine index. Taking into account all of the studied indices (not only LOF and COF), the best quality of clusters was shown by the CPCC (almost every time after removing outliers, its result is close to 100%) and the DBI index, they perform not much worse Silhouette index and Pseudo F index, so all of these indices are highly sensitive to outliers appearing in selected KBs. The nature of the data did not so much affect the quality of clusters after removing the outliers. When comparing different KBs containing rules, we may notice that, for most of them, when custom rules are removed, metrics have shown improvements in the range of 74% to 86% of cases, the exception is a small set of weather for which improvement was only achieved in about 62% of cases. Therefore, for the majority of the analyzed sets, in at least 75% of cases, the indicators showed an improvement in the quality of the resulting clusters after removing unusual rules. This confirms the consistency of the indices used, and it encourages the analysis of KBs in terms of outliers, as it is highly likely that we will improve their efficiency.

Research has also shown that there are indices that always show an improvement in the quality of the groups when removing a small number of outliers (1%), such as the Silhouette index, the DBI index, and the CPCC. It would seem that removing more outliers would have a better effect, but the fewer outliers found and removed, the more often most indexes indicated an improvement in quality. Most often, the quality of the groups improved after removing 1% or possibly 5% outliers, except for the Dunn index, which shows the opposite tendency. After a summary of the indications, it can be concluded that the removal of few of the most sensitive outliers is most effective in improving the quality of the clusters, and removing more and more of them (subsequent observations that are less and less outlier) is not so effective. Some of the indices may prove ineffective, for some data sets; therefore, the authors propose to use the indications of several indices and then synthesize their indications.

The statistical significance of differences between the methods of outlier detection and between the cluster quality measures was verified with the Chi square test while using the Statistica tool. There were 490 analyzed cases. It resulted from the following assumptions. For the analysis of statistical significance, the one optimal case (the best solution in terms of the cluster structure) was used. For each of the four methods of deviation detection and each of the seven measures of cluster quality and three options of the frequency (percentage) of the detected outliers (1%, 5%, 10%), as well as 6 KBs, an improvement/deterioration or no change in the quality of clusters was recorded after removing the rule that was indicated by the outlier detection method as unusual. At the level of statistical significance p<0.05, there were statistically significant differences between the cluster quality measures in the frequency of improvement/deterioration or no change in the cluster structure after the identification and omission of unusual rules. It was visible that most of the indices (Sillhouette, DBI, Gamma) in most cases (about 90%) responded with the improvement of the quality of clusters to the omission of unusual rules in the creation of rule clusters, while the Dunn index only in every fourth case positively responded to the omission of unusual rules in the cluster structure. There were no statistically significant differences (p>0.05) between the methods of outlier detecting: LOF, COF, *K*-MEANS, or SMALL
CLUSTERS in the frequency of improvement/deterioration or no change in the cluster structure. It turned out that all methods, similarly often, after removing the deviations found, recorded an improvement, deterioration, or no change in the cluster structure.

## 8. Conclusions

The aim of the paper is the analysis of using four outlier detection algorithms LOF, COF, *K*-MEANS, and SMALL
CLUSTERS in the context of rule-based KBs. Outliers in rules *If-Then* mean unusual rules, which are rare when comparing to others and should be explored by the domain expert as soon as possible. When there are a lot of rules *If-Then* in a given KB it is difficult to effectively manage them by domain experts or knowledge engineers. When the rules are clustered it is possible to short the time of the searching process and limiting itself to searching only the most appropriate cluster. Unfortunately, unusual rules may negatively affect the quality of the created clusters. Hence, the authors decided to check whether the quality of the clusters changes when unusual rules are omitted when creating clusters. Three methods are well known in the literature: LOF, COF, *K*-MEANS, but they were not used for rules *If-Then* before. The fourth method SMALL
CLUSTER is the method proposed by the authors. For different percentages (1%, 5%, 10%) of outliers and for seven different cluster quality indices (Silhouette, Dunn, DBI, Pseudo F, etc.), the frequency of improvement, deterioration, or no change in the quality of clusters after removing unusual rules was examined. The research shows that the most optimal results were obtained for two algorithms to detect outliers of LOF and COF. The SMALL
CLUSTERS algorithm proposed by the authors showed an improvement in the quality of the groups slightly worse (about 5% less cases).

The nature of the data did not so much affect the quality of clusters after removing the outliers. When comparing different KBs containing rules, we may notice that, for most of them, metrics have shown improvements when custom rules are removed.

At the level of statistical significance p<0.05, there were statistically significant differences between the cluster quality measures in the frequency of improvement/deterioration or no change in the cluster structure after the identification and omission of unusual rules. Most of the indices (Sillhouette, DBI, Gamma) responded with the improvement of the quality of clusters to the omission of unusual rules in the creation of rule clusters, while the Dunn index only in every fourth case positively responded to the omission of unusual rules in the cluster structure. There were no statistically significant differences (p>0.05) between the methods of outlier detecting: LOF, COF, *K*-MEANS, or SMALL
CLUSTERS in the frequency of improvement/deterioration or no change in the cluster structure. It turned out that all methods, similarly often, after removing the deviations found, recorded an improvement, deterioration, or no change in the cluster structure. In future research, we are going to examine the differences between these methods, including which rules indicated as unusual.

## Figures and Tables

**Figure 1 entropy-22-01096-f001:**
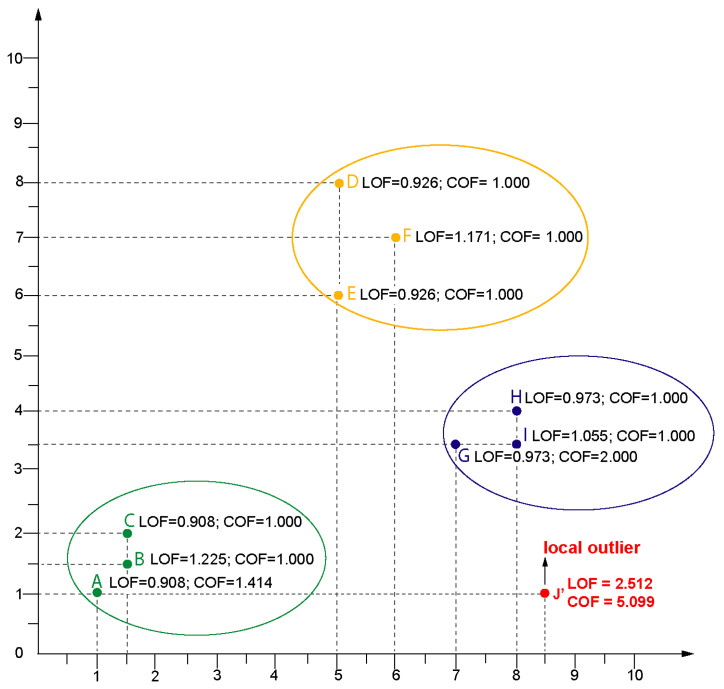
LOF and COF values for each data point.

**Figure 2 entropy-22-01096-f002:**
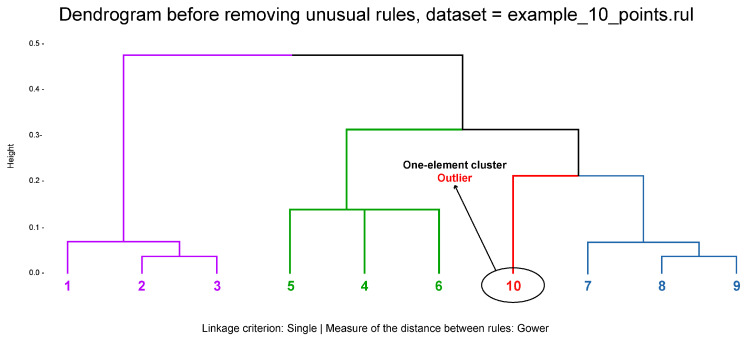
Dendrogram (four clusters) before removing unusual rules, SMALL
CLUSTERS algorithm, small one-element group in red.

**Figure 3 entropy-22-01096-f003:**
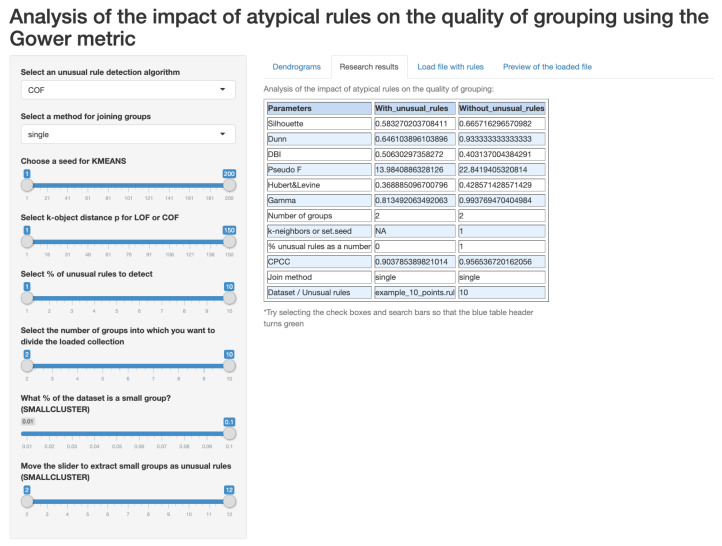
Application Rules Analysis—clustering and quality assessment.

**Figure 4 entropy-22-01096-f004:**
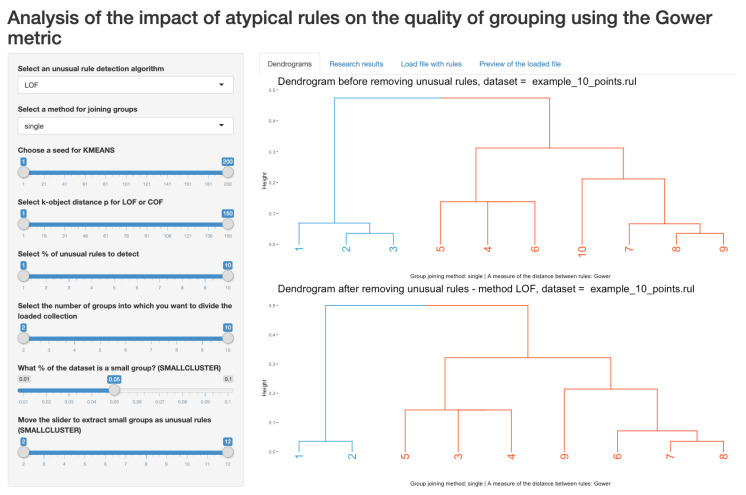
Application Rules Analysis—visualization of clusters.

**Figure 5 entropy-22-01096-f005:**
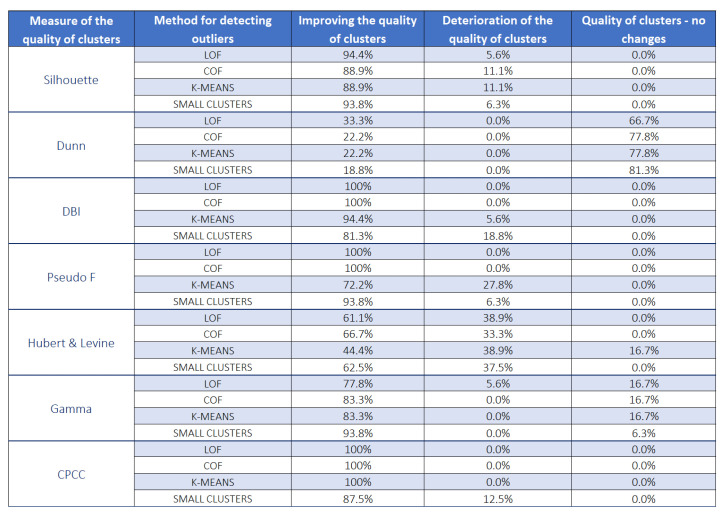
Cluster quality measurement vs. methods for detecting unusual rules and their influence on the frequency of improvement or deterioration of cluster quality.

**Figure 6 entropy-22-01096-f006:**
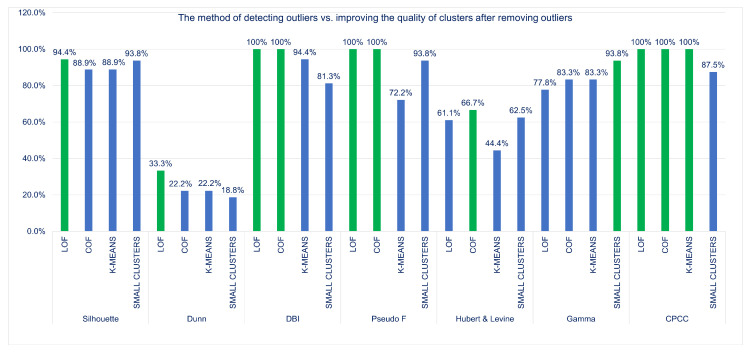
Cluster quality measurement vs. methods for detecting unusual rules and their influence on the frequency of improvement of cluster quality.

**Figure 7 entropy-22-01096-f007:**
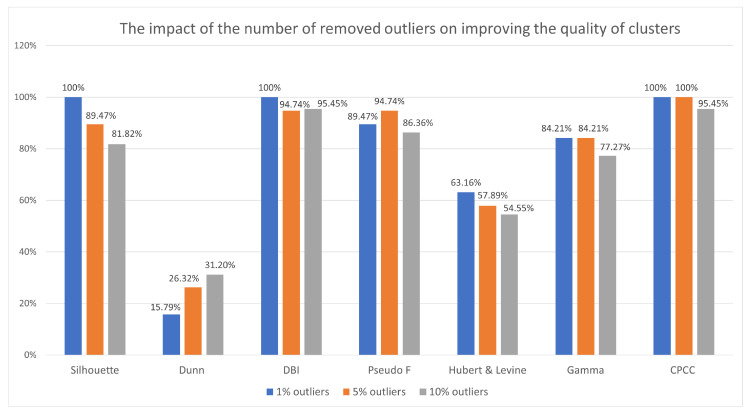
The impact of the number of removed outliers on improving the quality of clusters.

**Figure 8 entropy-22-01096-f008:**
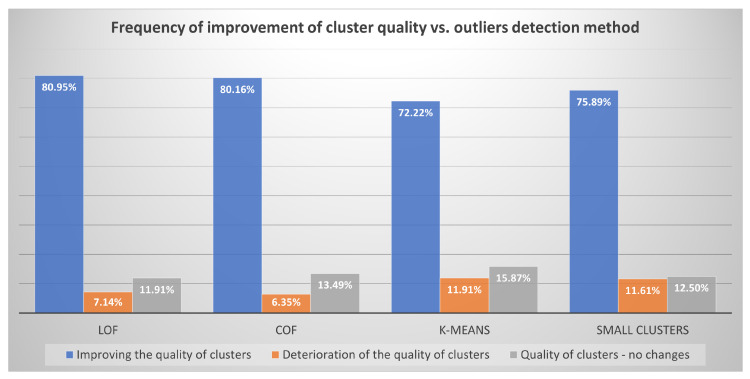
Frequency of improvement of cluster quality vs. outliers detection method.

**Figure 9 entropy-22-01096-f009:**
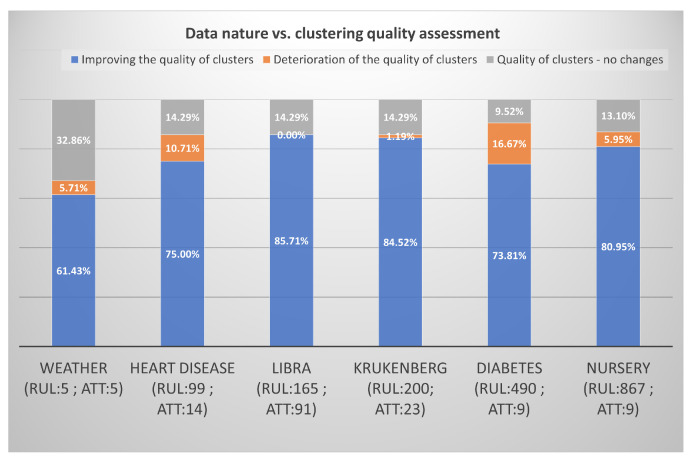
Data nature vs. clustering quality assessment—RUL (number of rules), ATT (number of attributes).

**Table 1 entropy-22-01096-t001:** The euclidean distance between each pair of data point.

	A	B	C	D	E	F	G	H	I
B	0.7071								
C	1.1180	0.5							
D	8.0622	7.3824	6.9462						
E	6.4031	5.7008	5.3150	2.0					
F	7.8102	7.1063	6.7268	1.4142	1.4142				
G	6.5	5.8523	5.7008	4.9244	3.2015	3.6400			
H	7.6157	6.9641	6.8007	5.0	3.6055	3.6055	1.1180		
I	7.4330	6.8007	6.6708	5.4083	3.9051	4.0311	1.0	0.5	
J′	7.5	7.0178	7.0710	7.8262	6.1032	6.5	2.9154	3.0413	2.5495

**Table 2 entropy-22-01096-t002:** Knowledge databases characteristics.

KB	#*Rules*	#*Attributes*	#*Items*
weather	5	5	14
heart disease	99	14	303
libra	165	91	360
krukenberg	200	23	111
diabetes	490	9	768
nursery	867	9	12,960

**Table 3 entropy-22-01096-t003:** Outliers in *heart disease*
*KB*—the case with 10% of outliers.

Rule Index	*LOF*	*COF*	*K*-*MEANS*	*SMALL CLUSTERS*
3	+			
4			+	
5			+	
6	+	+		+
9			+	
10	+	+	+	+
11	+	+	+	+
12			+	
16			+	
26			+	+
27			+	
31	+	+		
32			+	
42	+	+		+
47	+	+		
50	+	+		+
51	+	+		
55				+
62				+
68				+
86	+	+		+
94		+		

**Table 4 entropy-22-01096-t004:** Comparison of clustering quality indicators in terms of the frequency of showing improvement after removing outliers (unusual rules).

Silhouette	Dunn	DBI	Pseudo F	Hubert & Levine	Gamma	CPCC
90.43%	24.44%	96.73%	90.19%	58.53%	81.90%	98.48%

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
