# Peer review of "Exploration of Outliers in If-Then Rule-Based Knowledge Bases"

_entropy, 2020, doi:10.3390/e22101096_

Round 1

Reviewer 1 Report

Here are the comments for the authors.

  1. In abstract: There are many cluster algorithms used in numerous application. Author needs to explain clearly in abstract how their method is different and novel than others.
  2. Many techniques are used in many areas to support the decision support. Authors are advised to include those in their literature and explain it. Some examples are,
    a) Intelligent Predictive Decision Support System for Condition-Based Maintenance. https://link.springer.com/content/pdf/10.1007/s001700170173.pdf
    b) Data-Driven weather forecasting models performance comparison for improving offshore wind turbine availability and maintenance. https://doi.org/10.1049/iet-rpg.2019.0941
    c) A. Yahyaoui, A. Jamil, J. Rasheed and M. Yesiltepe, "A Decision Support System for Diabetes Prediction Using Machine Learning and Deep Learning Techniques," 2019 1st International Informatics and Software Engineering Conference (UBMYK), Ankara, Turkey, 2019, pp. 1-4, doi: 10.1109/UBMYK48245.2019.8965556
    d) H. Elaidi, Y. Elhaddar, Z. Benabbou and H. Abbar, "An idea of a clustering algorithm using support vector machines based on binary decision tree," 2018 International Conference on Intelligent Systems and Computer Vision (ISCV), Fez, 2018, pp. 1-5, doi: 10.1109/ISACV.2018.8354024.
    e) Ye Li, Ru-Po Yin, Yun-Ze Cai and Xiao-Ming Xu, "A new decision fusion method in support vector machine ensemble," 2005 International Conference on Machine Learning and Cybernetics, Guangzhou, China, 2005, pp. 3304-3308 Vol. 6, doi: 10.1109/ICMLC.2005.1527513.

  3.  

    Figure 1: Provide high-quality figure
  4. Fig 2: Graph looks confusing and of very low quality
  5. Fig 4: provide a high-quality figure

Author Response

Thank you for all valuable comments and tips. They certainly helped to improve the paper. They allowed us to look at our research from a different perspective. We hope we were able to respond to every comment and advice. We took into account all of them and added the necessary clarifications to the text of the manuscript.

Reviewer 2 Report

The major disadvantage of most of the grouping algorithms is their sensitivity to the presence of outliers, which strongly affects the deterioration of the resulting groups. An effective way of tackling this phenomenon is to isolate outliers in the initial stage (prepocessing) and thus prepare data for clustering.

The aim of the paper is the analysis of using four outlier detection algorithms LOF, COF, K-MEANS and SMALL CLUSTERS in the context of rule based knowledge bases. Outliers in rules IF - THEN mean unusual rules which are rare in comparing to others and should be explored by the domain expert as soon as possible. When there are a lot of rules IF - THEN in a given knowledge base it is difficult to effectively manage them by domain experts or knowledge engineers. When the rules are clustered it is possible to short the time of the searching process and limiting itself to  searching only the most appropriate cluster. Unusual rules may, unfortunately, negatively affect the quality of the created clusters.

In the article is missing an example of outliers (174, 196 and other) obtained for example from easy understanding first database.

(455 The wheather: the weather database, also known as tennis, is a simple set of data with decisions about playing tennis depending on the weather. The database consists of only 14 records. Each recorcontains weather information that can be useful for making decisions, the last column represents the decision class. Here we have qualitative and logical attributes.The independent variables are Outlook, Temperature, Humidity, and Wind. The dependent variable is whether to play tennis or not,).

The rest of databases examples are too specialized.for outliers example.

761 Not all abbreviations are icluded into list of  abbreviations. Please check it.

Author Response

(The authors gave the same response as above.)

Round 2

Reviewer 1 Report

Authors addressed all comments raised by reviewers and now ready for publication.